# Bright Yellow Luminescence from Mn^2+^-Doped Metastable Zinc Silicate Nanophosphor with Facile Preparation and Its Practical Application

**DOI:** 10.3390/nano14171395

**Published:** 2024-08-27

**Authors:** Mohammad M. Afandi, Sanghun Byeon, Taewook Kang, Hyeonwoo Kang, Jongsu Kim

**Affiliations:** 1Department of Display Science and Engineering, Pukyong National University, Busan 48513, Republic of Korea; andiafandi@pukyong.ac.kr (M.M.A.);; 2Electric Convergence Materials Division, Optics & Electronic Component Materials Center, Korea Institute of Ceramic Engineering and Technology, Jinju 52851, Republic of Korea

**Keywords:** metastable, zinc silicate, Mn^2+^ ions, luminescence, phosphor

## Abstract

Mn^2+^-doped β-Zn_2_SiO_4_, a metastable phase of zinc silicate, is widely acknowledged for the uncertainties linked to its crystal structure and challenging synthesis process along with its distinctive yellowish luminescence. In this study, a vivid yellow luminescence originating from Mn^2+^-doped metastable zinc silicate (BZSM) nanophosphor is suggested, achieved through a straightforward single-step annealing process. The reliable production of this phosphor necessitates substantial doping, surplus SiO_2_, a brief annealing duration, and prompt cooling. The verification of the phase is demonstrated based on its optical and crystallographic characteristics. Moreover, the effective utilization of excimer lamps in practical scenarios is effectively demonstrated as a result of the vacuum ultraviolet excitation property of BZSM nanophosphor. This outcome paves the way for additional deployment of metastable zinc silicate in various fields, consequently generating novel prospects for future advancements.

## 1. Introduction

Oxide materials have emerged as a promising class for luminescence applications due to their thermal stability, chemical inertness, and ability to incorporate various doping ions [1,2,3]. These materials are particularly advantageous in creating glass ceramic matrices, which are essential for developing efficient lighting devices. Additionally, doped metal oxide phosphors are highly favored for luminescence thermometry due to their low toxicity; high chemical, thermal, and photo-stability; and wide band gaps [4,5]. These phosphors, doped with lanthanides or transition metals, offer high sensitivity and resolution for non-contact thermal sensing, utilizing methods like luminescence intensity ratio (LIR) and rise/decay time thermometry [6]. The transition from sulfide-based phosphors to oxide-based phosphors in industrial displays highlights the enhanced effectiveness of oxide materials, which are currently prevalent in thin-film electroluminescent displays and other sectors [7]. In general, the varied structural and chemical characteristics of oxide materials render them extremely appropriate for a range of luminescent applications, spanning from illumination tools to sophisticated thermal detectors, such as the wide-bandgap Zn_2_SiO_4_.

Zinc silicate (Zn_2_SiO_4_) is a versatile material extensively studied for its luminescence properties, making it suitable for various applications, including phototherapy, dosimetry, and optoelectronics [8,9]. Zn_2_SiO_4_ is frequently classified into five distinct polymorphs, specifically referred to as α (I), β (II), III, IV, and V, each being associated with particular crystalline configurations denoted by *R*$\bar{3}$, *I*$\bar{42}$*d*, *P*2_1_/*n*, *Pcab*, and *Imma* space groups, respectively [10,11]. The luminescence of Zn_2_SiO_4_ can be significantly enhanced by doping with different ions. For instance, Gd^3+^ doping of Zn_2_SiO_4_ results in ultraviolet-B emission, which is beneficial for skin treatment and phototherapy applications [12]. Similarly, the incorporation of Mn^2+^, Yb^3+^, and Li^+^ ions in Zn_2_SiO_4_ enhances its persistent luminescence, making it a promising candidate for X-ray-induced photodynamic therapy (X-PDT) due to its ability to generate reactive oxygen species with reduced X-ray doses [13]. Eu^3+^-doped Zn_2_SiO_4_/ZnO composites exhibit red-shifted light emissions, making them potential candidates for optoelectronic applications [14]. The α-phase Mn^2+^-doped willemite stands out as the most remarkable optical functional material in Zn_2_SiO_4_, known for its efficient and economical production. This particular Mn^2+^-doped Zn_2_SiO_4_ demonstrates a prominent emission peak at 525 nm due to Mn^2+^ ion transitions and also displays ferromagnetic properties at reduced temperatures, enhancing its versatility [15,16]. Additionally, Mn doping has been reported to improve the dielectric properties in the Pb_1−y_La_y_Zr_x_Ti_1−x_O_3_ compound [17], demonstrating its effectiveness as a functional dopant in various solid-state applications. The manipulation of various dopants and synthesis techniques enables the adjustment of the luminescent characteristics of Zn_2_SiO_4_, rendering it a versatile material suitable for a wide range of sophisticated applications [18]. Nevertheless, the majority of studies have primarily concentrated on the α phase of Zn_2_SiO_4_, leaving a potential for enhancements in its β-phase variant, which is often overlooked due to its metastable nature.

β-Zn_2_SiO_4_, a metastable phase of zinc silicate, is commonly recognized for the ambiguities associated with its crystal structure, which results in it being less favored in comparison to the α-phase variant [19]. However, they have unique properties and potential applications in optoelectronics, especially Mn^2+^-doped β-Zn_2_SiO_4_. Mn^2+^-doped β-Zn_2_SiO_4_ has a broad emission spectrum, which is significantly influenced by the host lattice [20,21], making it a good candidate as a dopant in optical functional materials. Recent reports on the metastable β-phase Zn_2_SiO_4_:Mn^2+^ phosphor demonstrated yellow emission with a peak centered at 570 nm [22,23,24]. Kang et al. reported a critical stability of β-Zn_2_SiO_4_:Mn^2+^ with a 65% quantum efficiency under effective dopant concentration and synthesis parameters, attributed to its less crystalline structure and larger grain sizes, which contribute to reduced thermal quenching and longer decay times [25]. This β phase is embedded in an amorphous SiO_2_ with a rich stoichiometry. Moreover, β-Zn_2_SiO_4_ composites have shown promise in various applications, such as methylene blue dye degradation under ultraviolet radiation, although annealing above 700 °C can lead to the formation of a potential barrier due to the evolution of zinc silicate at the ZnO/β-SiC interface [26]. However, investigation of these luminescent materials is challenging because they are prepared with troublesome and sophisticated methods, such as smelting [25,27,28], morphology-controlled nanocrystals [29], and the formation of crystallized glass with specific compositions [30]. Thus, achieving continuous reproducibility is hindering the versatility of β-Zn_2_SiO_4_ practical applications.

In this work, we propose a promising bright yellow luminescence from Mn^2+^-doped metastable zinc silicate phosphor (β-Zn_2_SiO_4_:Mn^2+^, BZSM). The phosphor was prepared with a simple method following a one-step annealing process via solid-state reaction. The critical parameters to obtain reproducible BZSM nanophosphor were investigated. The optical and crystallography parameters were examined to clarify the existence of metastable zinc silicate in the developed phosphor. Furthermore, a practical application is presented through an Xe excimer lamp, which produces bright yellow light under a 19 kV (10 kHz) electrical source. The present investigation of BZSM nanophosphor offers a fresh and effective facile fabrication of metastable zinc silicate and lays the groundwork for the further versatile application of the β-Zn_2_SiO_4_ optical functional material.

## 2. Materials and Methods

### 2.1. Materials

Zinc acetate dihydrate [Zn(CH_3_COO)_2_·2H_2_O, 99.9%; Sigma Aldrich, St. Louis, MO, USA], manganese acetate tetrahydrate [Mn(Ch_3_COO)·4H_2_O, 99.99%; Sigma Aldrich], monoethanolamine (MEA, C_2_H_7_NO, 99%; Sigma Aldrich), isopropyl alcohol (IPA, C_3_H_8_O, 99.5%; Duksan Chemical, Ansan, Republic of Korea), and silicon dioxide with a 20 nm particle size (SiO_2_, 99%; CNVision, Rolling Meadows, IL, USA) were used.

### 2.2. Nanophosphor Fabrication

The fabrication of β-Zn_2_SiO_4_:Mn^2+^ (BZSM) nanophosphor is depicted in the flowchart shown in Appendix A. First, a 1 M ZnO:Mn solution is prepared by dissolving 1-*y* mol zinc acetate dihydrate, *y* mol manganese acetate tetrahydrate, and 1 mol monoethanolamine in 10 mL of IPA. The intermixture is then stirred vigorously with a magnetic stirrer at 1000 rpm and 50 °C for 4 h to obtain a transparent ZnO:Mn solution. The Mn content (*y*) ranged from 1 to 15 mol% in order to achieve optimal conditions. The solution was then dripped into the SiO_2_ powder to create a dough-like mixture of ZnO:Mn and SiO_2_. This mixture was prepared with various stoichiometric ratios, following the chemical parameters for the β-phase Zn_2_SiO_4_:2 of ZnO:Mn and 1 + *x* of SiO_2_, where *x* varies from −0.5 (indicating a deficiency of SiO_2_) to +1.0 (indicating an excess of SiO_2_) relative to the exact stoichiometry of *x* = 0 for Zn_2_SiO_4_. The dough was then left to dry for 6 h at room temperature to allow the organic solvent to evaporate. After drying, it was sieved using 1 mm and 75 µm meshes to retain some non-ground aggregates of the reagents. The powder was then calcinated at various temperatures of 700~1000 °C with various annealing durations from 3 to 20 min in a horizontal tube furnace, followed by rapid cooling to form the nanophosphors. After the annealing process, the nanophosphors formed not only as β-phase Zn_2_SiO_4_:Mn^2+^ but also as mixed-α-β-phase Zn_2_SiO_4_:Mn^2+^. These phases were distinguished by their unique emission colors under an ultraviolet lamp as well as their emission spectra.

According to our extensive efforts, the β phase was perfectly and brightly achieved only when the following four conditions were simultaneously satisfied: (1) heavy doping with *y*, Mn concentration = 5 mol%; (2) excess SiO_2_ (*x* ≥ 0.5) non-stoichiometry; (3) direct annealing at a temperature of 800 °C; and (4) rapid cooling at a rate of more than 400 °C/min. Direct annealing involves placing the sample directly into the furnace at a temperature of 800 °C for processing, while rapid cooling is accomplished by quickly extracting the sample from the well-insulated furnace and then blowing air on it with an electric fan. Careful attention was required to avoid fire hazards and thermal shock to the furnace elements. Additionally, since the cooling rate during natural cooling (power off) depends on the furnace’s insulation properties and varies across different furnaces, natural cooling is not recommended for achieving a controllable cooling rate. Therefore, the final BZSM phosphor was consistently obtained in the form of nanopowder. Finally, to demonstrate the metastability of the obtained BZSM nanophosphor, the sample underwent post-treatment with slow annealing at 1100 °C (heating rate of 5 °C/min) in an air atmosphere to facilitate the transformation to the α-phase counterpart.

### 2.3. Characterization and Instruments

Photoluminescence excitation (PLE) and photoluminescence emission (PL) spectra were recorded using a UV-Vis fluorescence spectrophotometer (F-7100, Hitachi, Tokyo, Japan). The vacuum UV PLE spectrum was measured with a diode array rapid analyzer system (Darsa Pro-5000, PSI, Suwon, Republic of Korea) under vacuum conditions of 10^−4^ Torr at room temperature. The crystal structures were analyzed by an X-ray diffractometer (XRD; PANalytical, Almelo, Netherlands). The XRD data were collected in the range of 10–80° in 2θ in a step-scan mode with a step size of 0.02° and a count time of 10 s per step. Particle morphology was examined using a field emission transmission electron (FE-TEM) microscope operated at 200 kV, which was equipped with an energy-dispersive X-ray spectrometer (EDS) for elemental mapping characterization.

## 3. Results and Discussion

### 3.1. Luminescence Characteristics

The distinctions between α- and β-phase Zn_2_SiO_4_:Mn^2+^ can be identified through their unique luminescent characteristics. Specifically, the α phase displays vibrant green light emission [31], while the β phase counterpart emits vivid yellow light [32]. Figure 1a shows the photoluminescence excitation (PLE) and photoluminescence emission (PL) of the β-Zn_2_SiO_4_:Mn^2+^ (BZSM) annealed at 800 °C with 5 mol% of Mn^2+^ concentration. Under the excitation wavelength (*λ*_ex_) of 265 nm, it can be observed that the sample demonstrates a spectrum that is quite extensive, showcasing a full width at half maximum (FWHM) of 54 nm, along with an emission peak that is distinctly positioned at 577 nm specifically within the yellow region. In detail, when the host lattice is excited by photons from the excitation wavelength, electron–hole pairs are formed at the Mn^2+^ luminescent centers. The recombination of these pairs results in the emission of yellow light. The genesis of this broad emission in the yellow spectrum can be ascribed optically to the spin-forbidden *d*-*d* transition that occurs within Mn^2+^ ions incorporated within the lattice structure of β-Zn_2_SiO_4_, as supported by various sources [25,27,32].

In general, the emission mechanism of the Mn^2+^ ion is associated with the radiative transition from the excited state of ^4^*T*_1_ to the ground state of ^6^*A*_1_. However, the spectral characteristics of Mn^2+^-doped Zn_2_SiO_4_ vary between the α and the β phase, in which α-Zn_2_SiO_4_:Mn^2+^ produces a green light emission with an emission peak centered at 525 nm, as presented in Appendix A. The luminescence of the Mn^2+^ ion spans a broad spectrum from blue-green to red and is heavily influenced by the host lattice and the Mn coordination environment [33,34,35]. The difference in energy levels of the Mn-O coordination between the α and β phases of Zn_2_SiO_4_ is likely the cause of the variations in the emission spectrum, as noted in earlier studies [27]. Additionally, under a monitoring wavelength (*λ*_em_) of 575 nm, our BZSM phosphor shows an excitation band spectrum with a peak at 265 nm, which is identified as the charge transfer band (CTB) of Mn^2+^ [36], as illustrated in Figure 1a. Notably, aside from the dominant CTB spectrum, several minor peaks can be observed, which are associated with the *d-d* excitation band of Mn^2+^ [23,36,37]. These specific excitation band peaks are at 356 nm [^4^*E* (^4^*D*)], 390 nm [^4^*T*_2_ (^4^*D*)], 420 nm [^4^*E* (^4^*G*)], 438 nm [^4^*T*_2_ (^4^*G*)], and 470 nm [^4^*T*_1_ (^4^*G*)].

The β-phase Zn_2_SiO_4_:Mn^2^ is well-known for its metastability compared to the α-phase counterpart. To demonstrate the metastability of our as-grown BZSM phosphor, the sample underwent post-treatment facilitating the phase transformation into the α phase. There was no noticeable difference in the body color of the samples before and after post-treatment. However, when irradiated with a 254 nm UV lamp, the post-treated sample emitted a bright green color, while the as-grown sample displayed a vibrant yellow color, as presented in the luminous photograph in the inset of Figure 1b. This change indicated the successful transformation of the BZSM phosphor into α-Zn_2_SiO_4_:Mn^2+^. Figure 1b provides a detailed illustration of the normalized PL spectra for the as-grown β-phase sample and the sample transformed into the α phase under 265 nm excitation. The transformed α phase exhibits green spectral characteristics with a peak centered at 525 nm, consistent with the previous literature on α-Zn_2_SiO_4_:Mn^2+^ phosphor [25,27,38,39] and our as-grown α phase, as shown in Appendix A. Furthermore, the PLE spectra of both the as-grown β phase and the transformed α phase nearly coincided, with the β-phase excitation being 3 nm red-shifted and broader, as shown in Appendix A. This suggests that the β phase is metastable and can transform into the stable α phase when provided with sufficient activation energy for the phase transformation. Here, we found that long annealing with slow cooling at temperatures above 900 °C (heating rate of 5 °C/min), identified in the previous literature [27,30,40] as the transition temperature, effectively induces the phase transformation. However, the transformed α phase cannot revert to the β phase, indicating the phase stability and/or irreversible nature of the transformation into the β phase.

Since the consistent reproducibility of β-phase Zn_2_SiO_4_ is affected by the synthesis conditions, it is essential to investigate the impact of annealing temperatures, annealing duration, and Mn^2+^ concentration. Herein, we investigated the best of the three parameters to produce consistent BZSM phosphors. Figure 2a displays the PL spectra of Zn_1_._95_SiO_4_:0.05Mn^2+^ at various annealing temperatures. Apart from the differences in intensity, a noticeable spectral variation is evident. This spectral variation and intensity could also be observed when the samples were excited with a 254 nm UV lamp, as shown in the luminous photographs in Figure 2b. The sample annealed at 850 °C exhibited the highest radiance power intensity, as shown in Figure 2c. Additionally, the color coordinates changed with the annealing temperature due to spectral variations, as detailed in Table 1. The alteration in the spectral characteristics can be attributed to the possibility that the samples did not grow in a single phase, meaning that the α-phase might have formed alongside the desired β-phase Zn_2_SiO_4_. The Gaussian deconvolution of the emission spectra, shown in Appendix A, was used to identify the fitting profiles of α- and β-Zn_2_SiO_4_:Mn^2+^ emissions. Even though the sample annealed at 850 °C had the most intense emission, the deconvolution graph reveals that it still exhibited minor α-phase emission. Conversely, the sample annealed at 800 °C showed practically non-existent α-phase emission, resulting in the best β/α emission ratio, as can be seen in Figure 2d. Therefore, it was deduced that the temperature of 800 °C represents the optimal annealing condition for achieving a uniform and uncontaminated BZSM sample.

The second parameter that needed to be optimized was the synthesis duration. A significant proportion of the α-phase Zn_2_SiO_4_ originates from a prolonged annealing process that lasts up to 4 h, giving enough time for the raw materials, ZnO and SiO_2_, to undergo complete reaction and yield crystallized Zn_2_SiO_4_. Conversely, the β-phase Zn_2_SiO_4_ tends to form as an amorphous phase as a result of the brief synthesis procedure. Therefore, a comprehensive analysis of the influence of annealing duration was essential to ensure the reproducibility of the newly formed BZSM nanophosphor. The PL spectra of Zn_1.95_SiO_4_:0.05Mn^2+^ following synthesis durations of 3, 5, 7, 10, 15, and 20 min at a temperature of 800 °C under ambient air conditions are illustrated in Figure 3a. The optimal duration for synthesizing consistently reliable results was identified as 10 min of annealing, these samples exhibiting a significant PL intensity alongside the highest β/α ratio, as shown in Figure 3b, which was obtained from Gaussian deconvolution of the PL spectra, referred to Appendix A. The phenomenon can be explained by noting that durations below 10 min led to decreased formation of Zn_2_SiO_4_, which was caused by inadequate energy for the reaction. On the contrary, samples synthesized for durations exceeding 10 min received excess energy, resulting in not only the formation of the metastable β phase but also the emergence of the stable α phase. Although the luminosity of those samples surpassed that of the specimens subjected to annealing for 10 min, the emission from the α phase coexisting with the β phase led to irregular outcomes. Consequently, it was deduced that a synthesis duration of 10 min represents the optimal condition for achieving consistent production of pure BZSM nanophosphor.

Our prior investigation demonstrated that the β phase is significantly influenced by the concentration of Mn^2+^, necessitating a substantial dopant of over 5 mol% [25]. Hence, in this study, we also analyzed the impact of Mn^2+^ concentration on the reliable reproducibility of BZSM nanophosphors. Figure 4a depicts the PL spectra of BZSM nanophosphor doped with Mn^2+^ contents of 1, 2, 3, 5, 10, and 15 mol%, annealed at 800 °C for 10 min in air. The PL intensity exhibits a rise as the quantity of Mn^2+^ rises, peaking at 3 mol%. This phenomenon can be clarified by the reinforcement of energy states and the increased presence of Mn^2+^ luminescent sites influenced by the charge carriers. However, the intensity decreases after surpassing the 5 mol% Mn^2+^ content threshold, eventually stabilizing at 5 mol%. The decline in PL intensity with increasing Mn^2+^ concentration can be elucidated by the widely acknowledged phenomenon of concentration quenching [41,42]. Upon reaching a specific threshold, such as 5 mol%, the doping level will diminish the inter-ionic (Mn^2+^-Mn^2+^) distance, resulting in the creation of Mn^2+^ ion pairs. This exchange interaction facilitates the migration of the excited electron between Mn^2+^ ions, consequently diminishing the emission efficiency. Subsequently, the PL intensity experiences a reduction beyond the 5 mol% Mn^2+^ concentration, as illustrated in the relative intensity plot included in Figure 4b. Furthermore, the sample containing 5 mol% Mn^2+^ annealed at 800 °C for 10 min demonstrates a notable and consistent emission of the β phase, which will serve as the primary focus throughout this study.

### 3.2. Crystallography Interpretation

To validate our inference regarding the Zn_2_SiO_4_ phase of the samples concerning their optical characteristics, an analysis of the crystal phase of the BZSM nanophosphor was conducted utilizing an X-ray diffraction (XRD) pattern. The XRD pattern of Figure 5a is presented to show the phase crystallography of the BZSM nanophosphor annealed at 800 °C. All the observed peaks are well-aligned with the standard JCPDF #14-0653, which was assigned as the β-phase Zn_2_SiO_4_. This result is consistent with the observation on the luminescence properties that the BZSM nanophosphor annealed at 800 °C formed as β-phase Zn_2_SiO_4_ without mixing with the α-phase counterpart. Pure β-Zn_2_SiO_4_ has not been assigned a crystal symmetry or space group, making it impossible to precisely identify the phase crystallography using Rietveld refinement. However, the lattice parameters can be estimated using the angular variation in the highest peak according to Bragg’s law. Accordingly, it has an orthorhombic crystal structure with lattice parameters of *a* = 8.4 Å, *b* = 5.1 Å, and *c* = 32.2 Å and a total volume of 1379.4 Å^3^, with *Z* = 16 and a cell density of 4.2 g/^3^, consistent with the previous literature on β-phase Zn_2_SiO_4_ [25,43].

All diffraction peaks displayed symmetric broadening uniformly, in contrast to the asymmetric broadening noted in specific crystal orientations with significant strain, a phenomenon often seen in nanoparticles [44,45]. This observation suggests the existence of BZSM phosphor at the nanoscale within the SiO_2_ particles, attributed to the brief annealing period impeding nucleation and growth. Moreover, a more extensive context was noted, identified as a locally epitaxially grown amorphous phase of excess SiO_2_, alongside the β phase serving as a crystal nucleus to reduce the local lattice mismatch. This phenomenon is attributable to the insufficient duration for the β phase to crystallize fully, given that the synthesis process is limited to a mere 10 min, resulting in the entrapment of the metastable BZSM within the SiO_2_ particles.

The BZSM sample, after undergoing transformation into its α-phase counterpart, was subjected to XRD analysis, with the corresponding pattern being illustrated in Figure 5b. The α phase resulting from the β-phase transformation exhibited a distinct and intense peak, in excellent agreement with the hexagonal crystal structure of α-Zn_2_SiO_4_ (JCPDF #37-1485). Unlike the β phase observed in conjunction with the amorphous SiO_2_, in addition to the α phase, the re-annealed specimen exhibited a prominent peak corresponding to cristobalite SiO_2_ (JCPDF #76-0938). The intense peak of cristobalite SiO_2_ is ascribed to the excess of SiO_2_ under stoichiometric conditions that underwent a thorough reaction throughout an extended annealing duration. The altered α-phase pattern displays a greater XRD intensity in contrast to the original β-phase pattern, suggesting a higher degree of crystallization in the α-phase sample, likely resulting from an adequate synthesis energy input. The transformation of the β phase appears to have been completed successfully. This outcome provides additional evidence supporting the metastable nature of β-phase Zn_2_SiO_4_:Mn^2+^ (BZSM), a characteristic that was assessed through both optical and crystallographic analyses.

Figure 6 shows transmission electron microscopy (TEM) images of BZSM nanophosphor to examine the particle morphology. The TEM image indicates that the phosphor particles are agglomerated and have irregular shapes, as illustrated in Figure 6a. Additionally, the particle size is approximately 30 nm in diameter, as shown in Figure 6b. This finding is consistent with the symmetric broadening of the diffraction peaks, confirming that the particles are on the nanoscale. The elemental mapping images from energy-dispersive X-ray spectroscopy (EDS) are illustrated in Figure 7. EDS analysis shows that Zn atoms are sporadically distributed within the SiO_2_ nanoparticles, suggesting that Zn_2_SiO_4_ crystal grains form in the outer areas of the SiO_2_. This can be explained by the short duration of the synthesis process, which limits nucleation to the outer regions of the SiO_2_ crystal grains. The low quality of Zn_2_SiO_4_ crystals can be attributed to insufficient time for the crystallization of Zn-Si-O, resulting in less condensed Zn atoms and larger grain sizes. This indicates inadequate diffusion reactions between Zn and Si necessary for forming Zn_2_SiO_4_. This finding aligns with the lower intensity observed in the XRD pattern shown in Figure 4, indicating reduced crystallization of the sample formed.

### 3.3. Practical Application

The α-Zn_2_SiO_4_:Mn^2+^ phosphor is frequently utilized in the plasma display panel (PDP) for its green-emitting characteristics in commercial applications. Numerous studies have highlighted the well-documented vacuum UV (VUV) excitation properties exhibited by this particular material, as evidenced in various research publications [46,47,48]. Hence, the VUV optical characteristics of the newly synthesized BZSM nanophosphor were further examined for potential practical use. Figure 8a illustrates the VUV excitation spectrum of the BZSM nanophosphor under 10^−4^ Torr vacuum conditions at room temperature. This spectrum exhibits a broad range of wavelengths from 135 to 175 nm, characterized by the presence of two distinct peaks. The prominent peak observed at 145 nm is ascribed to the excitation of the host lattice within the Zn_2_SiO_4_ matrix [38,49], while the secondary peak at 170 nm is linked to the 3*d*^4^4*s* transition of Mn^2+^ ions in solid-state materials [50]. The yellow emission of BZSM nanophosphor under VUV excitation can be understood by analyzing the underlying mechanism based on the findings presented above. Upon VUV excitation, the energy is absorbed by the host lattice and subsequently transferred to the Mn^2+^ ion, resulting in the emission of yellow light due to the radiative transition from the ^4^*T*_1_ excited state to the ^4^*A*_1_ ground state.

Figure 8b shows the PL spectra under excitation wavelengths of 142 nm and 172 nm. The observed spectra display a yellow emission featuring a peak wavelength of 577 nm. This peak aligns with the emission generated by 265 nm excitation, depicted in Figure 2, suggesting a reliable assignment of the emission mechanism. Furthermore, it is worth noting that the emission observed at a wavelength of 172 nm exhibits a significantly higher intensity compared to the emission observed at 142 nm. This notable discrepancy can be attributed to the intrinsic characteristics of the VUV excitation peak of BZSM nanophosphor, which is situated around the wavelength of approximately 140 nm, as depicted in Figure 8b.

The utilization of BZSM nanophosphor in practical scenarios is demonstrated. A layer of the phosphor was applied internally in a quartz tube through the utilization of a mixed solution containing 1 mL of ethyl acetate (C_4_H_8_O_2_, 99.5%; Daejung, Busan, Republic of Korea), 1 mL of butyl acetate (C_6_H_12_O_2_, 99.5%; Daejung), and 5 wt% of collodion (Kanto Chemical, Tokyo, Japan), with 1 wt% of Al_2_O_3_ (99.9%; Sigma Aldrich) as the binding agent. The binding agent was then mixed with 20 wt% of the BZSM phosphor powder. Following this process, the quartz tube with the coating was hermetically sealed under vacuum conditions with Xe gas at a pressure of 500 Torr to produce the BZSM Xe excimer lamp. Figure 8c demonstrates the emission spectrum of the apparatus under conditions of 19 kV and 19 kHz. The device produced a vivid yellowish tint, as depicted in the luminous image in the inset of Figure 8c, with the maximum emission wavelength positioned at 577 nm, underscoring the efficient application of the formulated BZSM phosphor. The emission light in excimer lamp applications currently exhibits poor homogeneity. We plan to improve lamp homogeneity through optimization of the coating process. In this work, we provide only a basic practical application of our proposed BZSM nanophosphor. We aim to further investigate and enhance its practical applications in future studies. Furthermore, the outcome of this study serves to broaden the scope of potential practical uses for excimer lamps, showcasing their versatility in various applications. It is our firm belief that the methodology we have pioneered in obtaining the yellow β-phase Zn_2_SiO_4_:Mn^2+^ holds significant promise in paving a clear path toward deeper explorations of this material. Furthermore, this innovative approach has the potential to significantly enhance the practical applications of such materials, thereby opening up new avenues for research and development in this field.

## 4. Conclusions

A successful demonstration was carried out on the bright yellow luminescence produced by metastable Mn^2+^-doped β-phase zinc silicate (BSZM) phosphor. The phosphor was synthesized using a simple one-step annealing process involving a solid-state reaction method. The essential factors for achieving consistent β-phase Zn_2_SiO_4_ were identified and validated through the analysis of optical and crystallographic properties. Additionally, the phosphor exists at the nanoscale, as evidenced by XRD and TEM data. Under vacuum ultraviolet (VUV) and UV excitation, the phosphor emitted a vibrant yellow light with an emission peak at 577 nm, which can be attributed optically to the spin-forbidden *d*-*d* transition of Mn^2+^ ions within the metastable host lattice. The application of this is exemplified through the utilization of BZSM phosphor in an Xe excimer lamp device. Operating under 19 kV and 19 kHz electrical conditions, it generates a bright yellow light that aligns with the results of photoluminescence analysis. We deduce that the yellow light produced by BZSM nanophosphor in a Xe excimer lamp device can be elucidated through an examination of the underlying mechanism driven by VUV excitation. The host material exhibits excitation characteristics that coincide with the Xe_2_^*^ emission (172 nm) and subsequently transfers to the Mn^2+^ ion, resulting in the yellow glow attributed to the radiative shift from the ^4^*T*_1_ excited state to the ^4^*A*_1_ ground state. The novel method for the practical utilization of the metastable β-phase Zn_2_SiO_4_ optical functional material is poised to facilitate diverse applications, thus creating new opportunities for exploration and advancement in this domain.

## Figures and Tables

**Figure 1 nanomaterials-14-01395-f001:**
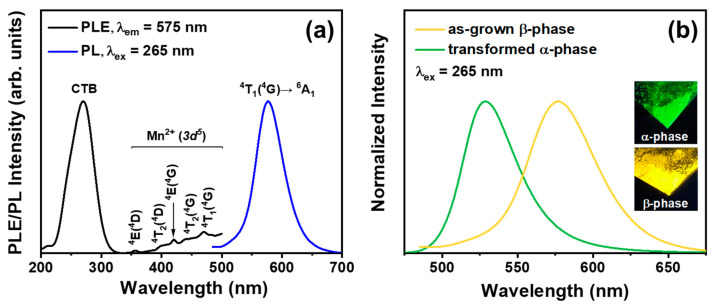
(**a**) Room temperature photoluminescence excitation (PLE) and photoluminescence emission (PL) of the BZSM nanophosphor annealed at 800 °C with 5 mol% Mn^2+^ concentration and (**b**) normalized PL intensity of as-grown β phase and transformed α phase of Zn_2_SiO_4_:Mn^2+^, with luminous photographs taken under a 254 nm UV lamp inset.

**Figure 2 nanomaterials-14-01395-f002:**
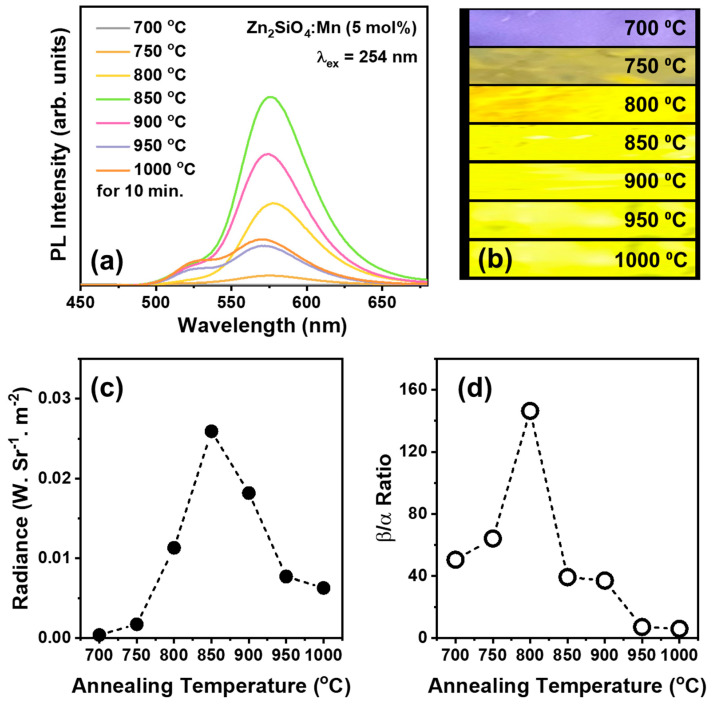
(**a**) Room-temperature PL spectra of the Zn_2_SiO_4_:Mn^2+^ with 5 mol% dopant concentration under various annealing temperatures for 10 min; (**b**) the respective photograph images of the samples under 254 nm lamp irradiation, taken with a Canon EOS RP (ISO 100, f/1.8, 1/240 s); and (**c**) radiance and (**d**) β/α emission ratios of the samples according to annealing temperatures.

**Figure 3 nanomaterials-14-01395-f003:**
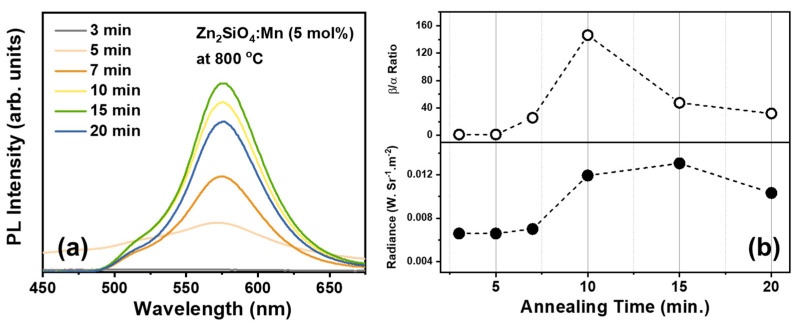
(**a**) Room-temperature PL spectra of the Zn_2_SiO_4_:Mn^2+^ with 5 mol% dopant concentration annealed at 800 °C for various synthesis times and (**b**) respective β/α emission ratios and radiances.

**Figure 4 nanomaterials-14-01395-f004:**
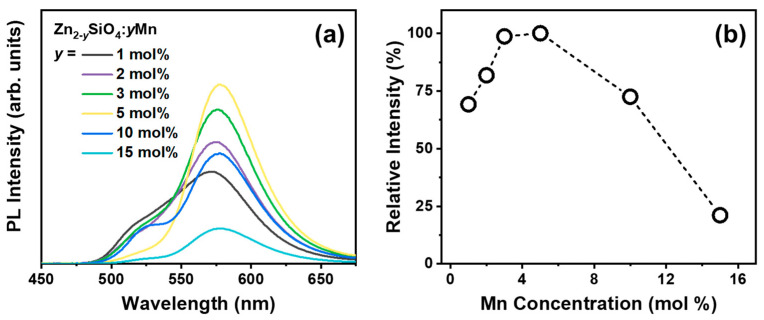
(**a**) Room-temperature PL spectra of Zn_2-y_SiO_4_:yMn^2+^ (y = 1, 2, 3, 5, 10, and 15 mol%) annealed at 800 °C for 10 min with (**b**) respective relative intensities.

**Figure 5 nanomaterials-14-01395-f005:**
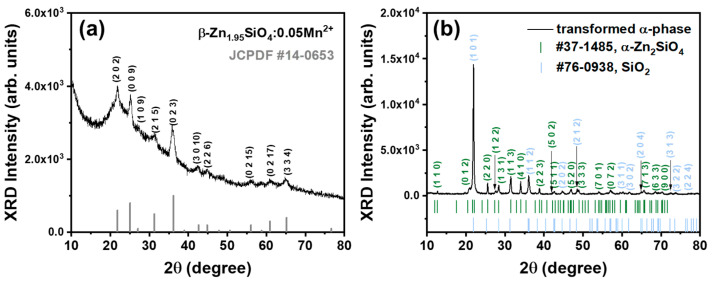
(**a**) X-ray diffraction (XRD) pattern of BZSM sample annealed at 800 °C with 5 mol% Mn^2+^ concentration and (**b**) XRD pattern of transformed sample re-annealed at 1000 °C for 4 h.

**Figure 6 nanomaterials-14-01395-f006:**
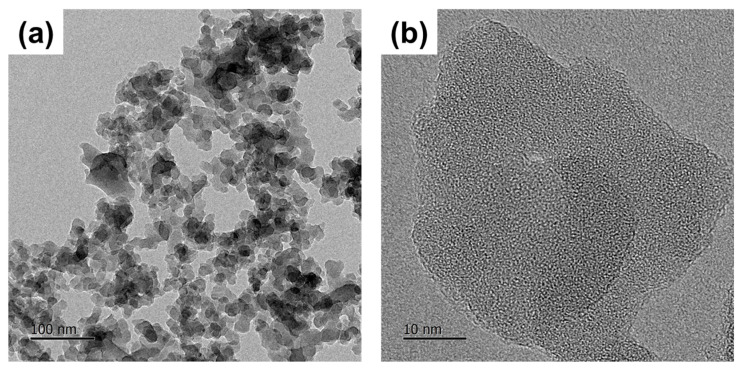
Transmission electron microscope (TEM) images of BZSM nanophosphor showing (**a**) the particles being agglomerated and (**b**) grain size.

**Figure 7 nanomaterials-14-01395-f007:**
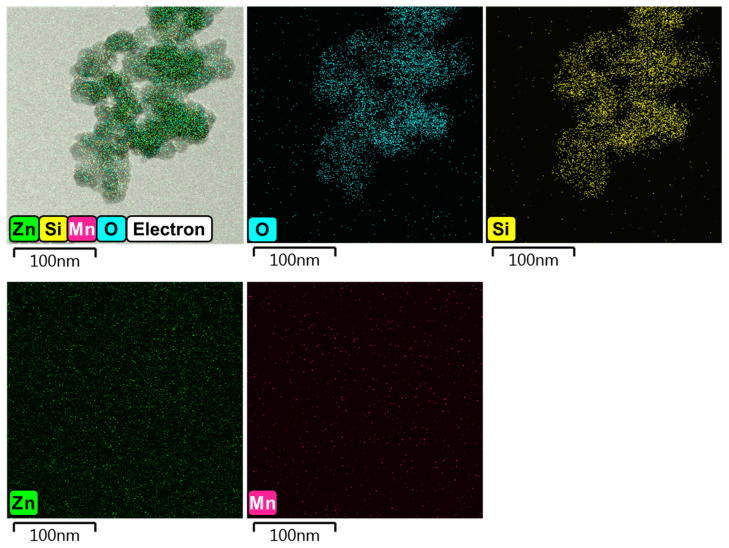
Energy-dispersive X-ray spectroscopy (EDS) images of BZSM nanophosphor showing the layered elemental image and elemental mapping images of O, Si, Zn, and Mn.

**Figure 8 nanomaterials-14-01395-f008:**
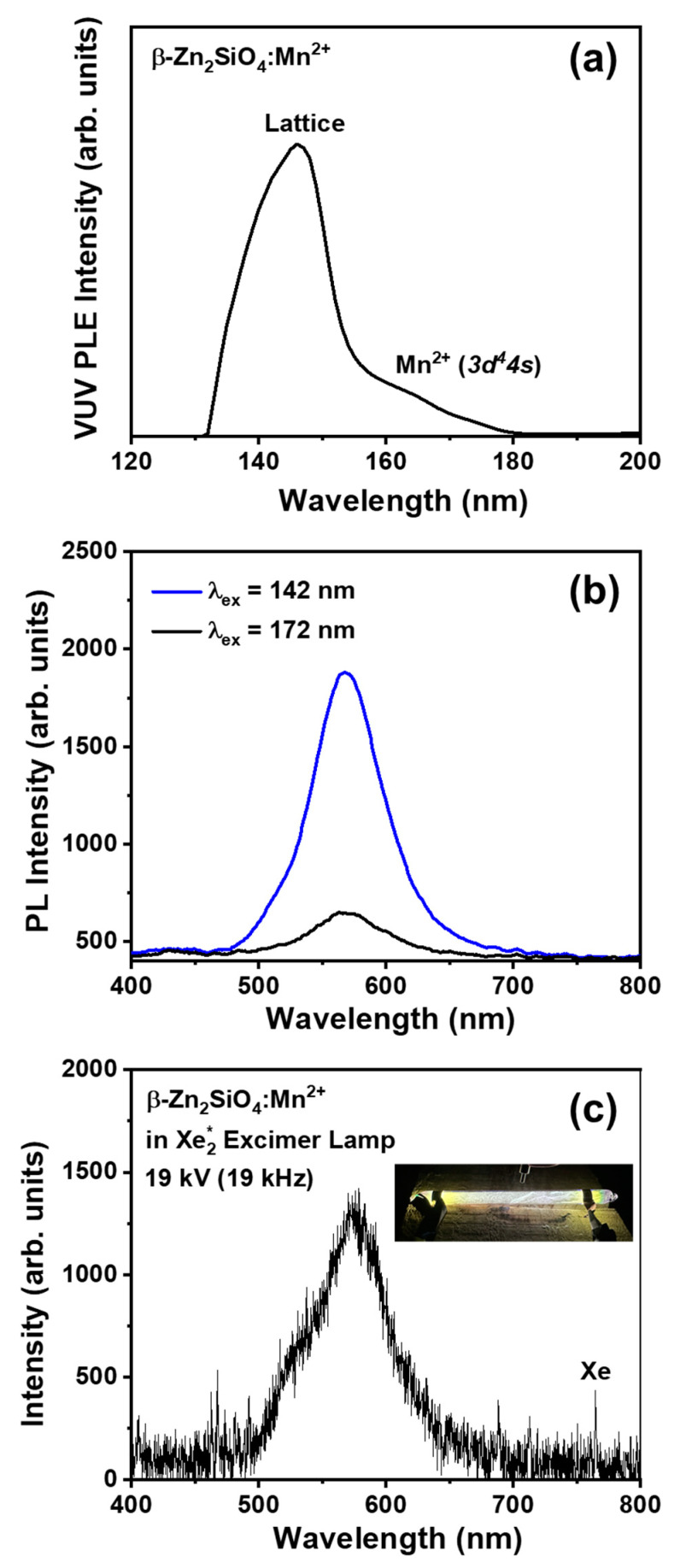
(**a**) Vacuum UV (VUV) PLE spectrum of the BZSM nanophosphor; (**b**) PL spectra of the BZSM nanophosphor under 142 and 172 nm VUV excitation; and (**c**) BZSM emission spectrum of the Xe_2_^*^ excimer lamp application, with a luminous photograph showing conditions of 19 kV and 19 kHz inset.

**Table 1 nanomaterials-14-01395-t001:** The optical parameters of Zn_2_SiO_4_:Mn^2+^ with 5 mol% Mn^2+^ concentration under various annealing temperatures.

Annealing Temperature (°C)	Emission Center (nm)	FWHM (nm)	Color Coordinate (x, y)
700	566	80.01	0.3259, 0.4032
750	574	62.75	0.4628, 0.5007
800	577	54.51	0.5097, 0.4849
850	575	53.94	0.4967, 0.4974
900	574	53.64	0.4849, 0.5081
950	570	59.00	0.4460, 0.5398
1000	569	78.65	0.4346, 0.5499

## Data Availability

Data will be made available on request.

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
