# Peer review of "Bright Yellow Luminescence from Mn^2+^-Doped Metastable Zinc Silicate Nanophosphor with Facile Preparation and Its Practical Application"

_nanomaterials, 2024, doi:10.3390/nano14171395_

Round 1
Reviewer 1 Report
Comments and Suggestions for Authors
The presented work describes and analyzes the spectroscopic properties of Mn2+-doped β-Zn2SiO4. In my opinion, there are some sections in the paper that raise doubts:
- What is the advantage of the presented work in comparison to the work of K. Omri and L. El Mir (10.1016/j.spmi.2014.02.022), which, by the way, is not referred to in this publication.
- Authors wrote that “After drying, it was sieved using 1 mm 109 and 75 μm meshes to remove unwanted residue and obtain the desired powder.” What residues do the authors expect, as all reagents were in the form of solutions or, in the case of SiO2, in powder form with a diameter of 20 nm? Sieving such powder, in my opinion, may just change the final stoichiometry of the powder, as the sieves may retain some non-ground aggregates of reagents.
- Authors describe “transformed α-phase of Zn2SiO4:Mn2” in the chapter 3.1.1. and Figure 2, but the description how it was prepared appear later, what introduces some misunderstanding. In my opinion, the description of the transformation of phase β into α should be described in the experimental part at the beginning of the publication.
- In my opinion Figure 2b is not necessary. All the transitions are described in the text.
- In the text authors write that samples were irradiated by 265 nm and in the figure 2c and figure caption is mentioned 254 nm. Which value is correct?
- “Furthermore, the PLE spectra of both the as-grown β-phase and 195 transformed α-phase nearly coincided, with the β-phase absorption being 3 nm red-196 shifted and broader, as shown in Figure S2, Supporting Information”. Authors should avoid using the terms excitation and absorption spectrum interchangeably, as both phenomena have different physical bases and should not be used interchangeably.
- The order of figures and tables is random, which makes it difficult to find a reference to them in the text. In my opinion, the description should come first, and then the figure/table. In the publication, Figure 3 appears first, then the description for Figure 2, then Figure 4 and Table 1 referring to Figure 3, and then the analysis of results for Figure 3. This introduces confusion in reading the publication and should be corrected.
- The results presented by the authors are not always consistent with each other. The authors describe that the annealing of samples at a temperature of 1100 oC shows the transformation of the β into α-phase (exhibiting green emission, Figure 2c), while for the samples annealed at 1000 oC the emission images show yellow emission (Figure 3b, and the figures in the supplementary). This leads to the conclusion that the formation of the α -phase is more related to the annealing time than to its temperature.
- The real content of Mn2+ in the n2-ySiO4:yMn2+ samples should be presented as it can be seen in Figure 5a that for 10 mol% Mn2+ is much more α-phase compared to the lower concentration of Mn2+
- The analysis of the XRD and especially the EDS maps didn’t convinced me that the authors have the right phase in the examined samples. The Si and O show distribution clearly assigned to the particles observed in SEM, but the Zn and Mn distribution is random, what suggest that the Mn2+ could exist in the SiO2 matrix, or that authors have composite of β−Zn2SiO4:Mn@SiO2.
In conclusion, the originality of the work should be emphasized more, and some of the results should be analyzed in greater detail. With these improvements, the work could be published.
Author Response
Reviewer #1
The presented work describes and analyzes the spectroscopic properties of Mn2+-doped β-Zn2SiO4. In my opinion, there are some sections in the paper that raise doubts:
|
|
Point-by-point response to Comments and Suggestions for Authors |
|
|
Comments 1: What is the advantage of the presented work in comparison to the work of K. Omri and L. El Mir (10.1016/j.spmi.2014.02.022), which, by the way, is not referred to in this publication |
|
|
Response 1: Thank you for your suggestion. The main advantages of our BZSM compared to the above-mentioned work are the less demanding synthesis conditions, as our preparation did not require a high annealing temperature and took only 10 minutes. In response to your suggestion, we have incorporated the mentioned work as a reference. |
|
|
Comments 2: Authors wrote that “After drying, it was sieved using 1 mm 109 and 75 μm meshes to remove unwanted residue and obtain the desired powder.” What residues do the authors expect, as all reagents were in the form of solutions or, in the case of SiO2, in powder form with a diameter of 20 nm? Sieving such powder, in my opinion, may just change the final stoichiometry of the powder, as the sieves may retain some non-ground aggregates of reagents. |
|
|
Response 2: Thank you for your attention to detail. We made a mistake concerning the sieving part. The correct answer aligns with your opinion that sieving was used to retain non-ground aggregates of reagents. In response to your question, we have corrected the sentence in the manuscript body, as highlighted in lines 113-114. |
|
|
Comments 3: Authors describe “transformed α-phase of Zn2SiO4:Mn2+” in the chapter 3.1.1. and Figure 2, but the description how it was prepared appear later, what introduces some misunderstanding. In my opinion, the description of the transformation of phase β into α should be described in the experimental part at the beginning of the publication. |
|
|
Response 3: Thank you for your suggestion. Accordingly, we have described the transformation of the β-phase to the α-phase counterpart in the experimental section as recommended. |
|
|
Comments 4: In my opinion Figure 2b is not necessary. All the transitions are described in the text. |
|
|
Response 4: We have removed Figure 2(b) as recommended. |
|
|
Comments 5: In the text authors write that samples were irradiated by 265 nm and in the figure 2c and figure caption is mentioned 254 nm. Which value is correct? |
|
|
Response 5: Thank you for your attention to detail. We acknowledge our mistake in the description of Figure 2(c), which is now Figure 1(b). The PL spectra are irradiated by 265 nm, while the luminous photograph was taken with the sample under a UV lamp (peaking at 254 nm). In response to your observation, we have corrected both the graph and the description in the main body. |
|
|
Comments 6: “Furthermore, the PLE spectra of both the as-grown β-phase and 195 transformed α-phase nearly coincided, with the β-phase absorption being 3 nm red-196 shifted and broader, as shown in Figure S2, Supporting Information”. Authors should avoid using the terms excitation and absorption spectrum interchangeably, as both phenomena have different physical bases and should not be used interchangeably. |
|
|
Response 6: Thank you for your correction. We have updated the terms of excitation in our manuscript accordingly. |
|
Comments 7: The order of figures and tables is random, which makes it difficult to find a reference to them in the text. In my opinion, the description should come first, and then the figure/table. In the publication, Figure 3 appears first, then the description for Figure 2, then Figure 4 and Table 1 referring to Figure 3, and then the analysis of results for Figure 3. This introduces confusion in reading the publication and should be corrected. |
|
|
Response 7: Thank you for your recommendation. We have rearranged the figure and table placements as suggested.. |
|
|
Comments 8: The results presented by the authors are not always consistent with each other. The authors describe that the annealing of samples at a temperature of 1100 oC shows the transformation of the β into α-phase (exhibiting green emission, Figure 2c), while for the samples annealed at 1000 oC the emission images show yellow emission (Figure 3b, and the figures in the supplementary). This leads to the conclusion that the formation of the α -phase is more related to the annealing time than to its temperature. |
|
|
Response 8: Thank you for your question. The luminous photograph of the 1100 ℃-annealed sample in Figure 3(b), now Figure 2(c), is bright yellow but shows the existence of a green spectrum. This sample was annealed for a short time, while the sample in Figure 2(c), now Figure 1(b), was annealed for a longer time. Indeed, annealing time is related to the formation of the α-phase. However, based on our extensive efforts, we cannot rule out the relation to annealing temperature. Therefore, we investigated the critical parameters to obtain a consistent β-phase, including annealing time, annealing temperature, and Mn2+ concentration. |
|
|
Comments 9: The real content of Mn2+ in the Zn2-ySiO4:yMn2+ samples should be presented as it can be seen in Figure 5a that for 10 mol% Mn2+ is much more α-phase compared to the lower concentration of Mn2+. |
|
|
Response 9: Thank you for your feedback. We acknowledge that the best condition to obtained pure β-phase is with maintaining the Mn2+ concentration at 5 mol%. Therefore, we are now mentioned the critical parameters to obtained consistent BZSM nanophosphor with 5 mol% of Mn2+ concentration in the experimental section. |
|
|
Comments 10: The analysis of the XRD and especially the EDS maps didn’t convinced me that the authors have the right phase in the examined samples. The Si and O show distribution clearly assigned to the particles observed in SEM, but the Zn and Mn distribution is random, what suggest that the Mn2+ could exist in the SiO2 matrix, or that authors have composite of β−Zn2SiO4:Mn@SiO2. |
|
|
Response 10: Thank you for your feedback. Indeed, the EDS elemental mapping images show that Zn and Mn are distributed randomly. We suspect that the Zn2SiO4 is growing in the outer areas of the SiO2 with lower crystal quality, which results in an uneven distribution. However, upon closer inspection, a more consistent distribution is observed in the outer areas with higher densities of Si and O. |
|

Reviewer 2 Report
Comments and Suggestions for Authors
The article is well-written, clearly outlining the operational conditions and offering an interesting study on manganese-doped zinc silicate. It highlights the advantages of oxide materials for luminescence applications, emphasizing their thermal and chemical stability. The study presents a detailed synthesis method for BZSM and investigates its optical and crystallographic properties. However, the absence of temperature-dependent X-ray diffraction analysis to confirm the photoluminescence findings is a notable drawback.
-Correct "velence band" in figure 2b
Comments on the Quality of English LanguageNo issues detected
Author Response
Reviewer #2
The article is well-written, clearly outlining the operational conditions and offering an interesting study on manganese-doped zinc silicate. It highlights the advantages of oxide materials for luminescence applications, emphasizing their thermal and chemical stability. The study presents a detailed synthesis method for BZSM and investigates its optical and crystallographic properties. However, the absence of temperature-dependent X-ray diffraction analysis to confirm the photoluminescence findings is a notable drawback:
|
Point-by-point response to Comments and Suggestions for Authors |
|
Comments 1: Correct "velence band" in figure 2b. |
|
Response 1: Thank you for your comments. We have included the XRD analysis for 800 °C and 1000 °C-annealing samples as a comparison and removed Figure 2(b) as suggested by Reviewer #1. |

Reviewer 3 Report
Comments and Suggestions for Authors
Referee report on “Bright yellow luminescence from Mn2+-doped metastable zinc silicate nanophosphor with facile preparation and its practical application”
This is quite interesting article, and it can be recommended for publication after clarifying some uncertainties.
1. The introduction clearly lacks information about Mn as a functional dopant of various solids developed for different applications. See, for example,
Dimza, Vilnis, et al. "Effects of Mn doping on dielectric properties of ferroelectric relaxor PLZT ceramics." Current Applied Physics 17.2 (2017): 169-173.
Luchechko, A., et al. "Luminescence Properties and Decay Kinetics of Mn and Eu Co-Dopant Ions in MgGa2O4 Ceramics." Latvian Journal of Physics and Technical Sciences 55.6 (2018): 43-51.
Novita, Mega, et al. "Study on Local-Structure Symmetrization of K2XF6 Crystals Doped with Mn4+ Ions by First-Principles Calculations." Materials 16.11 (2023): 4046.
2. A few important words need to be said about how luminescence depends on the impurity concentration. For example (line 65), the quoted 65% quantum efficiency is definitely dependent on the dopant concentration.
3. Fig. 2b. First of all, there is a typo (velence) !!!
Secondly, the drawing itself is incorrect. From which experimental data, it follows that the electron moves from the valence band to the conduction band ? Furthermore, such recombination of electron with Mn2+ will give the luminescence of Mn+ ions.
Note, that the 6A1 level is occupied, therefore the recombination shown in the figure is impossible.
4. When the host lattice is excited, electron-hole pairs are formed, but the role of holes is not discussed.
Author Response
Reviewer #3
Referee report on “Bright yellow luminescence from Mn2+-doped metastable zinc silicate nanophosphor with facile preparation and its practical application”
This is quite interesting article, and it can be recommended for publication after clarifying some uncertainties.
|
Point-by-point response to Comments and Suggestions for Authors |
|
Comments 1: The introduction clearly lacks information about Mn as a functional dopant of various solids developed for different applications. See, for example, Dimza, Vilnis, et al. "Effects of Mn doping on dielectric properties of ferroelectric relaxor PLZT ceramics." Current Applied Physics 17.2 (2017): 169-173. Luchechko, A., et al. "Luminescence Properties and Decay Kinetics of Mn and Eu Co-Dopant Ions in MgGa2O4 Ceramics." Latvian Journal of Physics and Technical Sciences 55.6 (2018): 43-51. Novita, Mega, et al. "Study on Local-Structure Symmetrization of K2XF6 Crystals Doped with Mn4+ Ions by First-Principles Calculations." Materials 16.11 (2023): 4046. |
|
Response 1: Thank you for your recommendation. We have incorporated the mentioned work as a reference for additional information about Mn as a functional dopant. |
|
Comments 2: A few important words need to be said about how luminescence depends on the impurity concentration. For example (line 65), the quoted 65% quantum efficiency is definitely dependent on the dopant concentration. |
|
Response 2: Thank you for your attention to detail. We added the condition to support the mentioned sentence in the manuscript body. |
|
Comments 3: Fig. 2b. First of all, there is a typo (velence) !!! Secondly, the drawing itself is incorrect. From which experimental data, it follows that the electron moves from the valence band to the conduction band? Furthermore, such recombination of electron with Mn2+ will give the luminescence of Mn+ ions. Note, that the 6A1 level is occupied, therefore the recombination shown in the figure is impossible. |
|
Response 3: Thank you for your correction. We have addressed the typo and misinterpretation accordingly. Additionally, we have removed Figure 2(b) as recommended by Reviewer #1, since the information is already described in the text. |
|
Comments 4: When the host lattice is excited, electron-hole pairs are formed, but the role of holes is not discussed. |
|
Response 4: Thank you for your feedback. We have provided a more detailed discussion of the light generation mechanism in our BZSM phosphor. |

Reviewer 4 Report
Comments and Suggestions for Authors
I read the article with interest and believe that after significant revision it can be considered for publication again.
Main questions to the authors and desirable improvements.
1. The article is poorly formatted.
Let's take lines 383-585 as an example.
Figure 9. (a) Vacuum UV (VUV) excitation spectrum of the BZSM nanophosphor, (b) VUV emission spectrum of the BZSM nanophosphor under 142 and 172 nm excitation, and (c) BZSM emission spectrum as the Xe2* excimer lamp application, with the inset is the luminous photograph under 19 kV. and 19 kHz.
Vacuum UV (VUV) excitation of what?
(b) VUV emission spectrum... Panel (b) shows spectrm in the visible spectral range.
Panel c. Inset demonstates poor homogeneity of the emission from the tbe. The spectrum is measured with high noise. It indicates the bad sensitivity of the easing equipment or weak luminescence.
2. Quite often, athors discss PL and EL the same way. This is not correct. EL is cathodolminescence or scintillation when doping ions are excited with participation of the condction and valence bands, whereas PL is excited via intracenter transitions.
3. To avoid an overloading of the text with technical details, I would suggest pt. Fig. 1 in the supplementary material.
4. Fig. 2b. You showed the creation of the unequilibrium electron (big vertical array). Where is the hole? What is its way in the figure?
5.Fig. 6.a. Why are XRD patterns so weak in comparison with the panel 6.b? Miller indexes should also be included in Panel 6.b.
6. Fig. 8: What is the sense of the practically black panel devoted to the dopant distribution?
Author Response
Reviewer #4
I read the article with interest and believe that after significant revision it can be considered for publication again. Main questions to the authors and desirable improvements.
|
Point-by-point response to Comments and Suggestions for Authors |
|
Comments 1: The article is poorly formatted. Let's take lines 383-585 as an example. Figure 9. (a) Vacuum UV (VUV) excitation spectrum of the BZSM nanophosphor, (b) VUV emission spectrum of the BZSM nanophosphor under 142 and 172 nm excitation, and (c) BZSM emission spectrum as the Xe2* excimer lamp application, with the inset is the luminous photograph under 19 kV. and 19 kHz. Vacuum UV (VUV) excitation of what? (b) VUV emission spectrum... Panel (b) shows spectrm in the visible spectral range. Panel c. Inset demonstates poor homogeneity of the emission from the tbe. The spectrum is measured with high noise. It indicates the bad sensitivity of the easing equipment or weak luminescence |
|
Response 1: Thank you for your comments. We have improved the article formatting as recommended and aligned it with the journal guidelines. We acknowledge that the caption for Figure 9, now Figure 8, was ambiguous. Therefore, we have revised the caption accordingly. In detail, panel A shows the VUV PLE spectrum of the BZSM nanophosphor, while panel B shows the PL spectra under VUV excitation. Indeed, the application of the excimer lamp has poor homogeneity. The optimization of the coating process for excimer lamp applications is ongoing and beyond the scope of this article. Here, we only provide a simple practical application of our proposed BZSM nanophosphor. We intend to further investigate and optimize its practicality in future work. |
|
Comments 2: Quite often, athors discss PL and EL the same way. This is not correct. EL is cathodolminescence or scintillation when doping ions are excited with participation of the condction and valence bands, whereas PL is excited via intracenter transitions. |
|
Response 2: Thank you for your feedback. We apologize for the incorrect interpretation. We have revised our manuscript to improve understanding and clarity accordingly. |
|
Comments 3: To avoid an overloading of the text with technical details, I would suggest pt. Fig. 1 in the supplementary material. |
|
Response 3: Thank you for your suggestion. We have moved the Figure 1 to the supplementary material as recommended. |
|
Comments 4: Fig. 2b. You showed the creation of the unequilibrium electron (big vertical array). Where is the hole? What is its way in the figure? |
|
Response 4: Thank you for your evaluation. We acknowledge that Figure 2(b) could lead to misunderstanding and misinterpretation. Consequently, we have removed it as recommended by Reviewer #1. |
|
Comments 5: Fig. 6.a. Why are XRD patterns so weak in comparison with the panel 6.b? Miller indexes should also be included in Panel 6.b. |
|
Response 5: The XRD patterns of the β-phase Zn2SiO4:Mn2+ are inferior compared to those of the transformed α-phase due to the lower quality of crystallization. The β-phase tends to grow amorphously, whereas the α-phase is well-crystallized. In our research, this can be attributed to the synthesis process, as the annealing time for the β-phase is only 10 minutes compared to 4 hours for the α-phase. In response to your suggestion, we have added the Miller indexes to the XRD pattern in Figure 6(b), now Figure 5(b). |
|
Comments 6: Fig. 8: What is the sense of the practically black panel devoted to the dopant distribution? |
|
Response 6: Thank you for your question. Indeed, the dopant distribution is somewhat unclear. However, we can observe spots with pink marks indicating the presence of the Mn element. Therefore, we intend to use the EDS elemental mapping images of Figure 8, now Figure 7, to elaborate on the existence of the dopant in our BZSM nanophosphor. |

Round 2
Reviewer 1 Report
Comments and Suggestions for Authors
The manuscript may be accepted in the present form
Reviewer 4 Report
Comments and Suggestions for Authors
Manuscript is improved and may be published in a present form.